# Delta Cord as a Radiological Localization Sign of Postoperative Adhesive Arachnoiditis: A Case Report and Literature Review

**DOI:** 10.3390/diagnostics13182942

**Published:** 2023-09-14

**Authors:** Yi-Ting Tu, Yung-Hsiao Chiang, Jiann-Her Lin

**Affiliations:** 1Department of Neurosurgery, Taipei Medical University Hospital, Taipei City 110301, Taiwan; mike100001048@gmail.com (Y.-T.T.); ychiang@tmu.edu.tw (Y.-H.C.); 2Taipei Neuroscience Institute, Taipei Medical University, Taipei City 110301, Taiwan; 3Division of Neurosurgery, Department of Surgery, School of Medicine, Taipei Medical University, Taipei City 110301, Taiwan

**Keywords:** adhesive arachnoiditis, postoperative, Chiari I malformation, syringomyelia, reoperation, neuroimaging

## Abstract

Postoperative adhesive arachnoiditis is an inflammatory response of the spinal leptomeninges that occurs after surgery and results in scar formation in the avascular nature of the arachnoid layer. Clinical manifestations of postoperative adhesive arachnoiditis include pain, sensory deficits, motor dysfunction, reflex abnormalities, and bladder or bowel impairment. In magnetic resonance imaging scans, signs of postoperative adhesive arachnoiditis can vary; however, some indicators can assist surgeons in locating the lesion accurately and, thus, in planning effective surgical interventions. This paper reports the case of a 37-year-old man with postoperative adhesive arachnoiditis after two surgeries for Chiari I malformation. This case illustrates the progressive development of the “delta cord sign”, which refers to the formation of a thick arachnoid band causing the spinal cord to adopt a triangular shape in the axial view. This phenomenon is accompanied by the sequential occurrence of syringomyelia. During intraoperative examination, we identified the presence of the delta cord sign, which had been formed by an arachnoid scar that tethered the dorsal spinal cord to the dura. This discovery enabled us to precisely pinpoint the location of the arachnoid scar and thus provided us with guidance that enabled us to avoid unnecessary exploration of unaffected structures during the procedure. Other localization signs were also reviewed.

## 1. Introduction

Postoperative adhesive arachnoiditis refers to an inflammatory response of the spinal leptomeninges following surgery. A previous study reported a prevalence of postoperative adhesive arachnoiditis of up to 20% in postoperative lumbar surgery patients [1]. The most common symptom of this response is pain (92.9% of cases), followed by motor deficit (35.7% of cases) and sensory deficit (28.6% of cases) [2]. Magnetic resonance imaging (MRI) is widely used to diagnose postoperative adhesive arachnoiditis. MRI findings associated with adhesive arachnoiditis include various features at the adhesive site, such as arachnoid cysts [3,4,5,6,7,8,9], clumped nerve roots [3,4,7,10], cord tethering [3,5,9,11,12,13], arachnoid septations [3,6,7,14,15], and arachnoiditis ossificans [3]. Additionally, nonspecific findings in cases of adhesive arachnoiditis may include hydrocephalus and syringomyelia [3,13]. These findings can be categorized as localization signs or as associated signs, with localization signs indicating the location of adhesive arachnoiditis and associated signs being consequences of adhesive arachnoiditis. In cases without localization signs before surgery, a diagnosis is typically established during surgical exploration [16]. However, surgical exploration of the spine is a time-consuming and invasive procedure. Precise localization of the adhesive site before surgery can enable surgeons to avoid unnecessary surgical exploration. Therefore, identifying the localization signs of adhesive arachnoiditis is crucial. This case report presents a new localization sign of adhesive arachnoiditis that is identifiable on MRI and that was confirmed by intraoperative findings for an adult patient with postoperative adhesive arachnoiditis. Additionally, this report reviews the literature regarding adhesive arachnoiditis and categorizes imaging findings as localization or associated signs.

## 2. Case Illustration

A 29-year-old man presented with bilateral hand numbness and clumsiness. Neurological examination revealed hyperreflexia in all limbs, indicating cervical myelopathy. MRI revealed a 5 mm descent of the tonsil below the foramen magnum with syrinx formation, which led to a diagnosis of Chiari I malformation. No congenital abnormalities were noted. The patient underwent a decompressive surgery involving suboccipital craniectomy, laminectomy of C1 and C2, and duroplasty. The first operation included a 2.5-centimeter-high and approximately 3-centimeter-wide suboccipital craniectomy. Intraoperatively, the arachnoid space was dissected to confirm cerebrospinal fluid (CSF) flow patency. After the surgery, the patient’s clumsiness subsided; however, poor wound healing and cerebral spinal fluid leakage occurred after 3 months, necessitating a revision surgery for wound repair.

Over an 8-year follow-up period, the patient’s MRI examinations revealed that the syrinx had initially subsided by the 3-month postoperative MRI but subsequently progressed over the next 5 years (Figure 1). In the 8th year of follow-up, the patient returned with worsening symptoms, including intermittent neck and back pain, progressive left hemiparesis, and paresthesia lasting 6 months. Initially, the patient could walk with a cane. However, his ambulation gradually deteriorated, and he required a quadricane for walking support. In addition, he experienced numbness, clumsiness, and loss of grasp strength. His modified Japanese Orthopedic Association (mJOA) scale score was 10, and his Barthel Index score was 50. Neurological examination revealed a muscle strength of 4 in the right upper and lower extremities and a muscle strength of 2 in the left upper and lower extremities. Atrophy of the intrinsic hand muscles was observed along with hyperreflexia in all four limbs. However, no Hoffman’s or Babinski signs were detected, and no cranial nerve involvement was noted. Cervical myelopathy was suspected, and cervical spine MRI confirmed the progression of the syrinx, that is, confirmed that the syrinx had extended to the C4 level, with edematous changes in the spinal cord. No brain or thoracolumbar abnormalities were discovered, and no epidural mass at the craniocervical junction was observed.

During a review of the patient’s cervical MRI series from previous years, a notable finding was made at the C2 level; that is, a thick arachnoid band was discovered to be attached to the spinal cord and the dural sac, causing the cord to take on a triangular shape in the axial view. This characteristic change in cord shape was termed the “delta cord” sign (Figure 2). Over time, progression of the syrinx from the C2 level to the C4 level was observed. After consideration of potential factors that could contribute to recurrent syringomyelia, postoperative adhesive arachnoiditis was suspected.

In order to address the postoperative adhesive arachnoiditis, a third surgery was performed. This surgery involved laminoplasty of C3–4 and extensive duroplasty from the suboccipital region to the C4 level. During the surgery, a thick arachnoid band was discovered. This band tethered the cord at the C2 level, confirming the presence of the delta cord sign. After separation of the arachnoid band, the syrinx collapsed, and the arachnoid space was no longer obstructed (Figure 3). After 1 year of postoperative follow-up, the syrinx collapsed, and an improvement of the edematous change in the spinal cord was observed (Figure 4). In addition, the patient’s functional status improved, with his mJOA score increasing from 10 to 11 and his Barthel Index score increasing from 50 to 60. Thus, he was able to walk with the assistance of a cane and perform the majority of his daily tasks, albeit with some dependency. He continued to experience residual paresthesia in his upper limbs and left lower limb.

## 3. Discussion

Adhesive arachnoiditis, or an arachnoid scar, is an inflammatory response of the arachnoid mater that occurs after spinal surgery, infection, subarachnoid hemorrhage, or foreign body reaction following the use of lumbar epidural injections, oil-based myelographic contrast agents, chemical irritation, or due to idiopathic causes [17,18,19]. Clinical manifestations of adhesive arachnoiditis include pain, sensory deficit, motor dysfunction, reflex abnormality, and bladder or bowel impairment [2,20]. However, the mechanism of postoperative adhesive arachnoiditis remains unclear. In one study, the inflammation process from cytokines in CSF inhibited the healing process and caused scar formation in the avascular nature of the arachnoid layer [17]. Adhesions or scarring within arachnoid channels may cause persistent syrinx or new onset symptoms without clear evidence of obstruction to CSF flow [21]. When the arachnoid scar blocked CSF flow, the pressure gradient caused syrinx formation [22]. Differences in microanatomical structures between the ventral and dorsal subarachnoid spaces were described in detail, and the dorsal site was reported to have more mesh-like arachnoid trabeculae than did the ventral site [23]. Koyanagi asserted that the dorsal subarachnoid space, which is rich in arachnoid trabeculae and veins, is prone to adhesive changes [19]. 

Since the 1970s, adhesive arachnoiditis has been a key focus of research on failed back surgery syndrome. Studies conducted in 1978 by Benner, Burton, Auld, and Quiles et al. used myelography to obtain evidence regarding adhesive arachnoiditis [24,25,26,27]. Subsequent research has analyzed imaging findings related to adhesive arachnoiditis. For example, Matsui et al. revealed cauda equina adhesion on MRI in their study involving a sample of patients with adhesive arachnoiditis [28]. Research indicates that T2 MRI with true fast imaging and steady-state precession or acquisition sequences can be used to identify and localize adhesive arachnoiditis through MRI [19,29]. The MRI signs associated with adhesive arachnoiditis encompass a range of features, including hydrocephalus, syringomyelia [3,7,13,14], arachnoid cysts [3,4,5,6,7,8,9], clumped nerve roots [3,4,7,10], cord tethering [3,5,9,11,12,13], arachnoid septations [3,6,7,14,15], and arachnoiditis ossificans [3]. These are categorized as localization signs, meaning that they indicate the location of adhesive arachnoiditis, or as associated signs, meaning that they represent the consequences of the condition (Table 1). In the case described in this report, the delta cord sign indicated the presence of an adhesive arachnoid scar at the C2 level, and the scar was later confirmed during surgery. The delta cord sign is unique in terms of its provision of diagnostic information from an axial view over the cord level, and it offers insights into the natural history of syrinx formation.

Typically, the arachnoid space is challenging to visualize on MRI because of its woven and loose tubular texture and because it is filled with CSF [23]. However, pathological changes can cause the arachnoid to become thicker and more compact, making it visible on MRI. In the present case, the arachnoid band connecting the spinal cord to the thecal sac initially caused cord distortion, which led to the formation of a triangular shape (Figure 2 and Figure 3). This separation of CSF flow from the cranial and caudal sides resulted in CSF filling the spinal canal and in eventual syrinx formation. Localization signs are valuable for neurosurgeons in that they enable precise surgical plans to be developed and unnecessary exploration of innocent structures to be avoided. Cord tethering, which is often revealed in sagittal views on MRI, is another localization sign [3,11,12,13]. In a Chiari I malformation case with postoperative pseudomeningocele and cord distortion, Belen et al. successfully detethered the spinal cord on the basis of a localization sign and repaired the pseudomeningocele [12]. Clumped root is another reported sign of lumbar adhesive arachnoiditis [3,11]. 

Surgical treatment options for postoperative adhesive arachnoiditis include adhesiolysis, expansile duroplasty, and CSF diversion from the ventricle or syrinx [19,21]. However, the outcomes of such treatments have been unsatisfactory in some cases [24,25,26,27]. For example, David et al. reported eight cases of craniocervical junction arachnoiditis with syringomyelia diagnosed through MRI. All eight cases exhibited clinical improvement after adhesiolysis and duroplasty during a mean follow-up period of 27 months (ranging from 10 to 60 months). Nevertheless, three of those cases required additional shunt surgery. Hirai et al. presented a case of adhesive arachnoiditis with a large arachnoid cyst after epidural injection administered during a cesarean section. The patient received adhesiolysis and arachnoid cyst diversion, and their symptoms improved over a period of 3 years [4]. In Koyanagi et al., only 30.8% of secondary adhesive arachnoiditis cases (*n* = 4) and all idiopathic adhesive arachnoiditis cases (*n* = 6) exhibited clinical improvement after surgical treatment [19]. However, unless localization is precise, an arachnoid scar may become widespread or not be identified during exploratory surgery [21]. Killeen et al. presented a case of severe adhesive arachnoiditis that developed after obstetric spinal anesthesia. The patient in question underwent several unsuccessful exploratory laminectomies and external drainage of the syrinx [19]. In the present case, the delta cord sign served as a localization sign indicating that the arachnoid band with cord tethering was located at the C2 level. The surgical strategy involved exploration at the C2 level to release the arachnoid band by using expansile duroplasty. Additionally, the surgical approach was extended to the C4 level and to even lower levels where necessary in order to assess the length of the arachnoid band and the flow of CSF intraoperatively. By using the delta cord sign as a guide, the surgical team was able to precisely localize the arachnoid scar, which facilitated the planning of a targeted and effective surgical intervention. In addition, extending exploration to adjacent levels enabled a comprehensive assessment of the condition of the area, which ensured that the optimal treatment for the patient was implemented. This approach was employed to alleviate the symptoms associated with adhesive arachnoiditis and to improve the patient’s overall clinical outcome.

We reviewed the potential causes of adhesive arachnoiditis in the present case. First, the treatment involved decompression surgery with duroplasty followed by a revision surgery for CSF leakage. Posterior fossa decompressive (PFD) craniectomy, laminectomy, and duroplasty are standard surgical treatments for symptomatic Chiari I malformation [30]. However, the appropriateness of routine use of duroplasty remains a topic of debate in the medical community. Yilmaz et al. indicated that patients for whom the cerebellar tonsil is positioned below the C1 arch may benefit from duroplasty [31]. However, a review by Xu et al. of 10 studies published from 2000 to 2017 revealed that PFD craniectomy with duroplasty led to more favorable clinical outcomes than did PFD craniectomy without duroplasty (mean difference = 0.85; 95% confidence interval [CI]: 0.73, 0.99; *p* < 0.05) [32]. However, PFD craniectomy with duroplasty was also associated with a high complication rate (mean difference = 0.34; 95% CI: 0.19, 0.60; *p* < 0.05) [33]; the most commonly observed complications were CSF leakage and arachnoid scar formation. The arachnoid scar formation was theoretically attributed to the exposure of the subarachnoid space to blood and muscle cell debris during the surgery as well as the use of dural grafts [33]. In the present case, the cerebellar tonsil was situated below the C1 level, prompting the surgical team to perform duroplasty to ensure sufficient decompression. However, subsequent occurrence of CSF leakage and the need for revision surgery resulted in repetitive trauma and a high likelihood of debris accumulation in the subarachnoid space. Accordingly, surgeons must carefully weigh the potential benefits and risks of duroplasty in each individual case and consider the patient’s specific anatomical characteristics and clinical condition. In summary, although duroplasty can improve clinical outcomes, it can also lead to an increased risk of complications such as CSF leakage and arachnoid scar formation.

Second, in the present case, we used fibrin glue to cover the suture site of the dural graft and to prevent CSF leakage. Such use of fibrin glue is common practice in surgical procedures [34,35,36]; however, fibrin glue is used to mimic the final step in the coagulation pathway to form a fibrin clot [35], which can cause a hypersensitivity reaction, as was previously observed [37]. Alternatively, fibrin glue may be inadvertently squeezed into the subarachnoid space during wound closure because it is inserted between the muscular and fascial layers. Hayashi et al. reported a case of fibrin glue-induced adhesive arachnoiditis during the treatment of a sacral meningeal cyst; in that case, direct exposure to fibrin glue caused adhesion processes to occur within the subarachnoid space [38]. In the present case, exposure to fibrin glue also caused an adhesive formation process within the subarachnoid space.

Third, dural graft material is a risk factor of adhesive arachnoiditis [33]. We used DuroGen (Integra, Saint Priest, France) artificial dural graft for duroplasty in the first and revision surgeries. As an allogenic substitute, dural graft material can stimulate the inflammatory process, especially when in direct contact with the subarachnoid space [39].

Finally, surgery itself and blood content are also risk factors of adhesive arachnoiditis [17,18,24,25,26,27,28,40,41]. Thus, avoiding unnecessary or repetitive exploration, removing blood clots during dura closure, employing water-tight-fashion closure, and preventing fibrin glue injection into the subdural space could be effective methods for preventing adhesive arachnoiditis.

The exact underlying mechanisms of adhesive arachnoiditis remain uncertain. Although multiple case series have shed light on the progression of arachnoid scar development, a well-established animal model has yet to be investigated. Over the preceding decade, attempts to establish animal models that closely resemble spinal arachnoiditis have focused primarily on pain assessment and the development of syringomyelia [42,43]. However, the complex structure of the subarachnoid space and trabeculae pose challenges to replicating all of the intricacies of this region [44]. Due to the infrequency of adhesive arachnoiditis occurring and variations in the condition among individuals, treatments for the condition vary considerably. This variability makes conducting clinical trials or establishing treatment guidelines or algorithms difficult; decisions are often made on the basis of medical professionals’ experience or the results of case series or small group studies. Notably, no comprehensive review with image-based categorization for adhesive arachnoiditis has been conducted. The present study reviewed original articles related to adhesive arachnoiditis published between 2012 to 2022. A total of 10 articles were included in this review, and the articles comprised case reports and case series. The articles contained a combined total of 18 reported cases of adhesive arachnoiditis. The average age of the affected individuals was 42.8 years, with the age range spanning 27 to 66 years. The review indicated that women (12 cases) were more susceptible to adhesive arachnoiditis than were men (5 cases). In the review, the potential causes or etiologies of adhesive arachnoiditis were diverse and included the following factors: post-spinal surgery (2 cases), subarachnoid hemorrhage (3 cases), inherited genetic anomaly (6 cases), epidural anesthesia (2 cases), spinal injury (1 case), and infection (2 cases). A noteworthy finding from the review was a report by Pasoglou et al. that the largest observed series of adhesive arachnoiditis cases occurred within a family group (*n* = 6); however, that study did not identify any genetic abnormalities associated with adhesive arachnoiditis, and the study lacked imaging information and treatment details [6].

In recent years, surgical approaches have shifted toward releasing arachnoid scars or cysts instead of diverting CSF. Of the 10 cases identified in this review of patients who underwent decompressive surgery, 7 involved adhesiolysis or cyst aspiration, 1 involved duroplasty, and 2 involved additional shunting surgery. However, in three cases, detailed surgical techniques were not provided. Notably, some cases in the reviewed articles described laminectomy alone without mentioning adhesiolysis or cyst aspiration. However, in our experience, performing laminectomy alone without releasing the arachnoid scar is not common practice.

The focus of the present study was imaging examinations and their correlations with surgical outcomes. Among the patients in the reviewed articles of this study, 12 exhibited localization signs, including arachnoid cysts (*n* = 8), arachnoid septations (*n* = 3), tethered or distorted cords (*n* = 4), and clumped nerve roots or cauda equine (*n* = 4). Additionally, three patients had syringomyelia as an associated sign, with this sign often occurring in combination with other localization signs. Notably, the use of localization signs has improved surgical precision and is associated with favorable functional outcomes [21,37,38,39,40,41,42,43], except in cases of diffuse lesions or those involving other etiologies like infection or inflammation [3,39,42]. Within the surgical intervention groups of the reviewed studies (*n* = 13), 11 patients experienced improved (*n* = 5, 38.4%) or stabilized (*n* = 6, 46.1%) clinical or radiological outcomes, whereas 2 cases (15.3%) exhibited worsened conditions. In the conservative treatment groups of these studies (*n* = 3), one case (33.3%) worsened, and two cases (66.7%) remained stable. Regarding the cases with worsened outcomes, Killeen et al. reported a patient with diffuse lesions who underwent multiple surgical interventions without a satisfactory outcome [11]. Pasoglou et al. reported a case (Case 1) with poor outcomes after surgery; however, no details regarding the imaging or surgical techniques were reported [6]. Jurga et al. reported a case (Case 2) of a poor outcome after surgery; however, the study noted unknown inflammatory disease in its pathology report, and an image of adhesive arachnoiditis included in their article indicated that the adhesive arachnoiditis was not as typical as other cases of the condition are [14]. Overall, surgical treatment generally provides an opportunity to improve or maintain functional outcomes. Research findings on the topic are compiled in Table 2 [4,5,6,7,8,9,10,11,14,15]. We conducted an organized review of these articles to offer guidance to physicians who encounter similar cases in the future.

## 4. Conclusions

Postoperative adhesive arachnoiditis refers to an inflammatory response of the spinal leptomeninges following surgery. Recognizing the localization signs of adhesive arachnoiditis is crucial to effective surgery planning and to avoiding unnecessary exploration of innocent structures.

In this study, we introduced the delta cord sign, a unique indicator that provides diagnostic information from an axial view over the cord level and that offers insights into the natural history of syrinx formation. Releasing the fibrotic arachnoid band by using the delta cord sign was able to restore CSF flow and led to functional recovery in the present case.

## Figures and Tables

**Figure 1 diagnostics-13-02942-f001:**
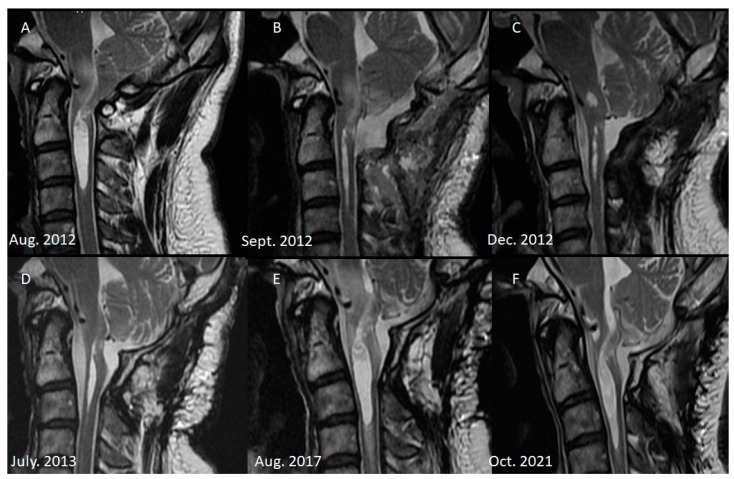
Series of cervical MRI. (**A**) Typical Chiari I malformation before decompressive surgery. (**B**) Due to transient dysphagia, we arranged a C-spine MRI 1 week after decompressive surgery. Mild edematous change over the medulla was observed, and short-term steroid administration was prescribed. Adequate decompression was achieved, and the syrinx shrunk. (**C**) CSF leakage was noted 3 months after the first surgery, for which dura repair was performed. MRI before the surgery revealed no signs of infection, and the syrinx length had decreased. (**D**) Six months after the revision surgery, MRI revealed mild progression of the syrinx and that the cord to the thecal sac was tethered. (**E**) In the 5th year of follow-up, the syrinx continued to progress and extended to the C3 level. Edematous change to the spinal cord was also observed. However, no clinical symptoms had developed. (**F**) In the 8th year of follow-up, progressive left side hemiparesis was observed, and MRI revealed syrinx extension to the C4 level, for which revision surgery was performed.

**Figure 2 diagnostics-13-02942-f002:**
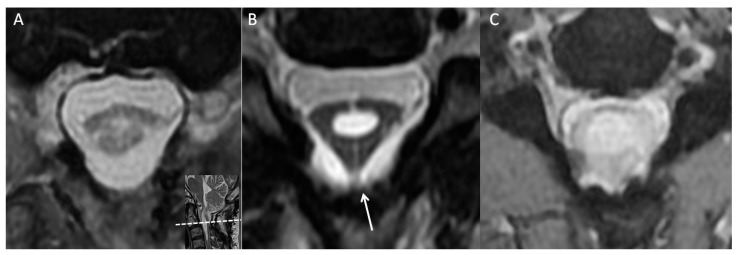
Axial view of the C2 level at multiple time points. (**A**) Three months after the first surgery. (**B**) After 1 year, a thick arachnoid band (white arrow) had attached to the posterior aspect of the thecal sac, giving the cord a triangular shape, that is, the “delta cord” sign. Syrinx progression was observed from that point onward. (**C**) The spinal cord had attached to the thecal sac, and syrinx progression and cord edematous change were observed.

**Figure 3 diagnostics-13-02942-f003:**
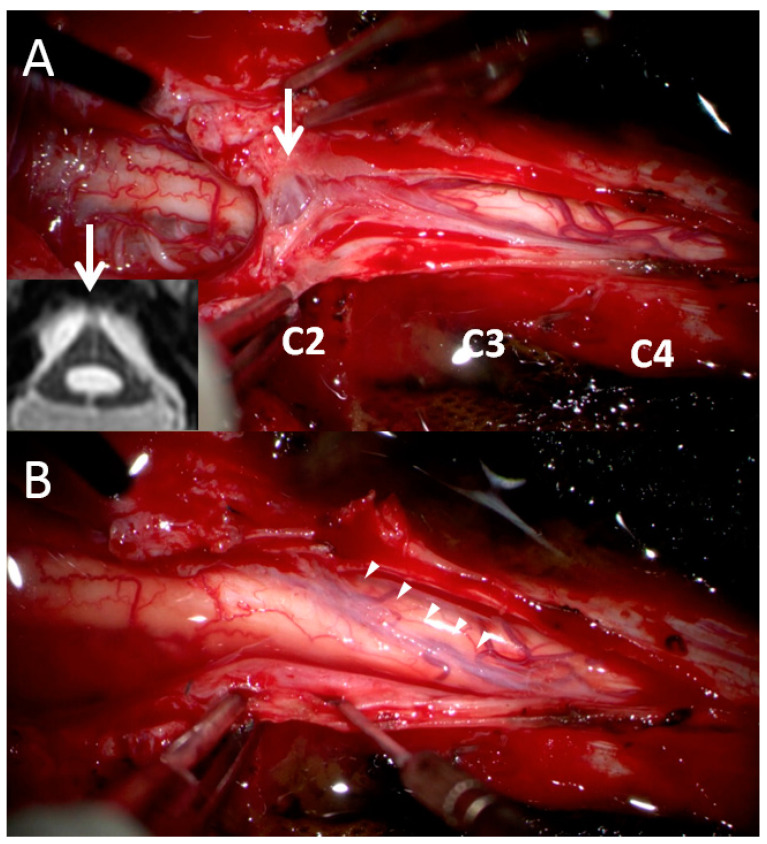
Surgical findings. (**A**) At the C2 level, a thick arachnoid band (white arrow) extended to the C4 level and obstructed CSF flow between the cranial and caudal sites. This finding was consistent with our previous MRI finding of the delta cord sign (we rotated the axial MRI view 180° to match the surgical finding). (**B**) After the release of the arachnoid band (arrow heads), the cord collapsed and connected CSF flow from the cranial and caudal sites.

**Figure 4 diagnostics-13-02942-f004:**
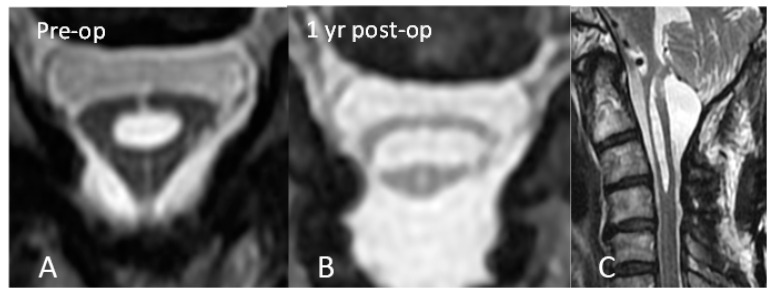
(**A**) Delta cord sign and (**B**) 1 year after surgery. The shape of the cord became more oval. (**C**) Sagittal view 1 year after surgery. Decreasing diameter of the syrinx and absence of edematous change were observed.

**Table 1 diagnostics-13-02942-t001:** Localization and associated signs of adhesive arachnoiditis.

Computed Tomography or MRI Finding	Evidence Type	Hypothesis of Pathophysiology
Nerve root clumping, cauda equina adhesion	Localization sign	Formation of arachnoid scar in accordance with the surgery and wound healing [28]
“Delta cord” sign, disrupted cord contour	Localization sign	Fibrinous bands lead to adhesions between the cord of the thecal sac [11]
Arachnoid septations, arachnoid cysts	Localization sign	Arachnoid web formation by chronic inflammation of the arachnoid [3]
Syringomyelia, syrinx	Associated sign	Blocked CSF flow likely caused by adhesions or pressure gradient at the obstruction site [3]
Pseudomyelomeningocele	Associated sign	Continuous CSF pressure that forces CSF to come into contact with muscular layer [12]

**Table 2 diagnostics-13-02942-t002:** Summary of case reports or series published from 2012 to 2022.

Studies	Characteristics	Suspected Etiology	Location and Image Sign Type	Management	Outcomes
Killeen. 2012 [11]	27 y/o, F	Noxious agent (chlorhexidine) or blood in the pia-arachnoid	Multilevel clumped nerve roots and disrupted cord contour (localization signs)	Surgery (laminectomy, adhesiolysis, and shunting)	Worsened: Unable to work and dependent
Ishizaka. 2012 [5]	66 y/o, F	Subarachnoid hemorrhage	Deformity of cord and arachnoid cyst (localization signs)	Surgery (laminectomy)	Improved: Can walk unaided but with abdominal paresthesia
Hirai. 2012 [4]	29 y/o, F	Insertion of epidural tube or puncture, intradural administration of bupivacaine, or epidural bleeding	Arachnoid cyst compressing the spinal cord, and convergence of cauda equine (localization signs)	Surgery (laminectomy and adhesiolysis)	Improved: Can walk without cane but with numbness, moderate lower limb weakness, and anuresis
Pasoglou. 2014 [6]	Case 1: 35 y/o, F	Rare inherited genetic anomaly (not confirmed)	No image shown	Surgery (laminectomy)	Short amelioration then worsening
	Case 2: 50 y/o, M	same as above	No data	Surgery *	Death due to perioperative complications
	Case 3: no data	same as above	No data	No data	No data
	Case 4: 49 y/o, F	same as above	No image shown	Surgery *	Regression of syrinx; no results regarding functional outcomes
	Case 5: 45 y/o, F	same as above	No image shown	Surgery *	No results regarding functional outcomes
	Case 6: 49 y/o, M	same as above	Arachnoid septations and cysts (localization signs)	Surgery (laminectomy and adhesiolysis)	Regression of syrinx; no results regarding functional outcomes
Carlswärd. 2015 [10]	29 y/o, F	Epidural blood patch with subsequent inflammation	Lumbar clumped root (localization sign)	Conservative	Worsened and wheelchair dependent
Todeschi, 2017 [8]	57 y/o, F	SAH of ruptured aneurysm from Adamkiewicz artery	Arachnoid cysts (localization sign)	Surgery (laminectomy and adhesiolysis)	Improved: Can walk with crutches
Maenhoudt. 2018 [7]	Case 1: 59 y/o, F	Meningitis	Arachnoid cysts (localization sign)	Surgery (laminectomy and cyst aspiration)	No improvement of neurological status
	Case 2: 50 y/o, F	SAH of ruptured PICA aneurysm	Arachnoid cysts, septations, clumped cauda equine and syringomyelia (localization and associated signs)	Surgery (laminectomy and cyst aspiration)	Improved, including syrinx regression and gait and bladder function improvements
Kleindienst. 2020 [14]	59 y/o, M	Spinal injury	Arachnoid septations and syringomyelia (localization and associated signs)	Surgery (laminectomy and duroplasty)	Stabilized neurological function
Jurga. 2021 [9]	Case 1: 50 y/o, F	Post-spinal surgery or spinal block,	Spinal sac and spinal cord deformities (localization signs)	Conservative	Unimproved and wheelchair dependent
	Case 2: 26 y/o, M	Unknown inflammatory disease of the spinal cord	Intrathecal cyst and distorted spinal cord (localization signs)	Surgery *	Unimproved and wheelchair dependent
	Case 3: no age record, F	Post-spinal surgery	Archnoid cysts (localization sign)	Conservative	Unimproved and wheelchair dependent
Safi. 2021 [15]	29 y/o, M	Staphylococcus cohnii infection	Syringomyelia and arachnoid septations (localization and associated signs)	Surgery (laminectomy, adhesiolysis, and shunting)	Improved: Fully independent and syrinx regression
Present case	29 y/o, M	Repeated spinal surgery, CSF leakage, foreign material use (Fibrin glue and artificial dural graft)	Delta cord sign, cord tethering, and syringomyelia (localization and associated signs)	Surgery (laminectomy, adhesiolysis, and duroplasty)	Improved: Can walk with cane

* No details regarding surgical techniques were provided; F: female, M: male, PICA: posterior inferior cerebellar artery, SAH: subarachnoid hemorrhage.

## Data Availability

All data are available in this article.

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
