# Peer review of "Delta Cord as a Radiological Localization Sign of Postoperative Adhesive Arachnoiditis: A Case Report and Literature Review"

_diagnostics, 2023, doi:10.3390/diagnostics13182942_

Round 1

Reviewer 1 Report

1. Reframe table no. 1

2. Conclusion should be more descriptive. It is in just 2 lines

3. Author can cite more new references

4. In discussion section, please discuss more about available studies and then discuss your result in accordance with it

NO

Author Response

Dear Reviewer:

Thank you for your valuable comments and suggestions. My replies for your comments and suggestion point by point as below:

  1. Reframe table no. 1

We reframed the table no. 1 included table size, capitalization and made it more organized.

  1. Conclusion should be more descriptive. It is in just 2 lines

         We edited the conclusion with more detailed descriptions.

  1. Author can cite more new references

         We cited more new reference from recent studies and highlighted it.

Page 1: ”arachnoid cyst [4,21,37-39,41,42], clumped nerve roots [4,21,39,44], cord tethering [3-6,37,42], arachnoid septations [4,38-40,43], and arachnoiditis ossificans [4].”

Page 5: “The MRI signs associated with adhesive arachnoiditis encompass a range of features, such as hydrocephalus, syringomyelia [4,6,39,40], arachnoid cyst [4,21,37-39,41,42], clumped nerve roots [4,21,39,44], cord tethering [3-6,37,42], arachnoid septations [4,38-40,43], and arachnoiditis ossificans [4].”

  1. In discussion section, please discuss more about available studies and then discuss your result in accordance with it

         We have compiled recent articles on adhesive arachnoiditis and created a table that outlines case characteristics, suspected underlying causes, imaging findings categorized based on localization/associated signs, treatment approaches, and outcomes. We believe this review will offer guidance to physicians when encountering similar cases in the future.

  We appreciate your thorough review of the manuscript and for highlighting its shortcomings. Besides, we sought for a professional English edition service for our manuscript. Our goal was to ensure the article is comprehensive and includes the latest research findings.

Best regards.

Reviewer 2 Report

This is the case report describing a 37-year-old male with postoperative adhesive arachnoiditis after two surgeries for Chiari I malformation, demonstrating that the gradually forming the delta cord sign with sequential syringomyelia. During the intraoperative examination, authors identified the presence of the delta cord sign, which was formed by an arachnoid scar that was tethering the dorsal spinal cord to the dura. This finding led authors to pinpointing the location of the arachnoid scar, thereby providing a guidance to avoid any unnecessary exploration of innocent structures during the procedure. 

I have no major concerns except the following suggestions:

Fig. 3 caption….”compactible à compatible”

Fig. 4 there is no “A , B, C” marked on top of each image from the left to the right, although written in the figure legend. Please add “A B C” on top of each brain scan image.

Table 1 Not organized properly; narrow down the width…(such a poorly represented table overlapping with line number…)

Conclusion: There are unnecessary periods at the end of sentence: please remove.

Please read your manuscript carefully and correct several grammatical and formatting errors. 

Author Response

Dear Reviewer:

   Thank you for your valuable comments and suggestions. My replies for your comments and suggestion point by point as below

  1. Fig. 3 caption….”compactible à compatible”

        We fixed the spelling error and highlighted the new word.

  1. Fig. 4 there is no “A , B, C” marked on top of each image from the left to the right, although written in the figure legend. Please add “A B C” on top of each brain scan image.

        We fixed the error of this picture.

  1. Table 1 Not organized properly; narrow down the width…(such a poorly represented table overlapping with line number…)

         We reframed the table no. 1 included table size, capitalization and made it more organized.

  1. Conclusion: There are unnecessary periods at the end of sentence: please remove.

        We edited the conclusion with more detailed descriptions and corrected the grammatical error.

   We appreciate your thorough review of the manuscript and for highlighting its shortcomings. Besides, we sought for a professional English edition service for our manuscript. Our goal was to ensure the article is comprehensive and includes the latest research findings.

Best regards.

Reviewer 3 Report

The authors reported the case who demonstrated the delta  cord sign suggesting arachnoid adhesion after posterior decompression for Chiari malformation associated with syringomyelia. And They reviewed literatures about clinical feature, radiological findings and pathogenesis in arachnoid adhesion. The delta cord sign is not newly information as arachnoid adhesion. It is difficult to understand the discussion. The authors should summarize the reviews (discussion) to understand easily, dividing items.

   I am interested in this article as the review of arachnoid adhesion. There is no systematic review about arachnoid adhesion.  The authors were encourage to review clinical feature, radiological findings, pathogenesis, surgical treatment and its outcome at detail, using tables, figures and shames. 

Author Response

Dear Reviewer:

  Thank you for your valuable comments and suggestions. My replies for your comments and suggestion point by point as below:

"The authors reported ...  I am interested in this article as the review of arachnoid adhesion. There is no systematic review about arachnoid adhesion. The authors were encourage to review clinical feature, radiological findings, pathogenesis, surgical treatment and its outcome at detail, using tables, figures and shames."

    We have compiled recent articles on adhesive arachnoiditis and created a table that outlines case characteristics, suspected underlying causes, imaging findings categorized based on localization/associated signs, treatment approaches, and outcomes. We believe this review will offer guidance to physicians when encountering similar cases in the future.

  We appreciate your thorough review of the manuscript and for highlighting its shortcomings. Besides, we sought for a professional English edition service for our manuscript. Our goal was to ensure the article is comprehensive and includes the latest research findings.

Best regards.

Round 2

Reviewer 3 Report

I recommend to accept this article. The authors properly revised this article, according to the suggestions of he reviewers. I think that this article became to be the valuable review about arachnoid adhesion.